# Walking Speed is the Sole Determinant Criterion of Sarcopenia of Mild Cognitive Impairment in Japanese Elderly Patients with Type 2 Diabetes Mellitus

**DOI:** 10.3390/jcm9072133

**Published:** 2020-07-06

**Authors:** Noritaka Machii, Akihiro Kudo, Haruka Saito, Hayato Tanabe, Mariko Iwasaki, Hiroyuki Hirai, Hiroaki Masuzaki, Michio Shimabukuro

**Affiliations:** 1Department of Diabetes, Endocrinology and Metabolism, Fukushima Medical University, Fukushima 960-1295, Japan; noritaka@fmu.ac.jp (N.M.); a-kudoh@fmu.ac.jp (A.K.); saito-h@fmu.ac.jp (H.S.); htanabe@fmu.ac.jp (H.T); mariko99@fmu.ac.jp (M.I.); hiroyuki@fmu.ac.jp (H.H.); 2Division of Endocrinology, Diabetes and Metabolism, Hematology, Rheumatology (Second Department of Internal Medicine), University of the Ryukyus, Okinawa 903-0215, Japan; hiroaki@med.u-ryukyu.ac.jp

**Keywords:** type 2 diabetes mellitus, walking speed, sarcopenia

## Abstract

Diabetes mellitus is a risk factor for mild cognitive impairment (MCI) and dementia. However, how the clinical characteristics of MCI patients with type 2 diabetes mellitus are linked to sarcopenia and/or its criteria remain to be elucidated. Japanese patients with type 2 diabetes mellitus were categorized into the MCI group for MoCA-J (the Japanese version of the Montreal cognitive assessment) score <26, and into the non-MCI group for MoCA-J ≥26. Sarcopenia was defined by a low skeletal mass index along with low muscle strength (handgrip strength) or low physical performance (walking speed <1.0 m/s). Univariate and multivariate-adjusted odds ratio models were used to determine the independent contributors for MoCA-J <26. Among 438 participants, 221 (50.5%) and 217 (49.5%) comprised the non-MCI and MCI groups, respectively. In the MCI group, age (61 ± 12 vs. 71 ± 10 years, *p* < 0.01) and duration of diabetes mellitus (14 ± 9 vs. 17 ± 9 years, *p* < 0.01) were higher than those in the non-MCI group. Patients in the MCI group exhibited lower hand grip strength, walking speed, and skeletal mass index, but higher prevalence of sarcopenia. Only walking speed (rather than muscle loss or muscle weakness) was found to be an independent determinant of MCI after adjusting for multiple factors, such as age, gender, body mass index (BMI), duration of diabetes mellitus, hypertension, dyslipidemia, smoking, drinking, estimated glomerular filtration rate (eGFR), HbA1c, and history of coronary heart diseases and stroke. In subgroup analysis, a group consisting of male patients aged ≥65 years, with BMI <25, showed a significant OR for walking speed. This study showed that slow walking speed is a sole determinant criterion of sarcopenia of MCI in patients with type 2 diabetes mellitus. It was suggested that walking speed is an important factor in the prediction and prevention of MCI development in patients with diabetes mellitus.

## 1. Introduction

Dementia refers to a condition in which cognitive function, which has reached a normal level, is sustainably reduced due to acquired brain damage, thereby interfering with daily life and social life (International Classification of Diseases (ICD11)) [1], Diagnostic & Statistical Manual of Mental Disorders, 5th ed. (DSM-5) [2]). There are several conditions that resemble cognitive decline or prodromal symptoms of dementia but cannot be diagnosed as dementia. One such condition, mild cognitive impairment (MCI), was proposed by Flicker et al. [3] and established by Petersen et al. [4]. It has been previously reported that more than 50% of individuals with MCI later develop dementia [4]. Modifiable risk factors for mild dementia include (middle-aged) hypertension, diabetes mellitus, (middle-aged) obesity, dyslipidemia, smoking, physical activity, and depression [5,6].

Diabetes duration is a risk factor for MCI and dementia (Alzheimer’s dementia, vascular dementia, and mixed dementia) [7,8,9]. In the Japanese population, the risk of developing Alzheimer’s disease and vascular dementia is approximately twice as high in patients with type 2 diabetes mellitus as it is in healthy individuals [10]. The risk factors for MCI in patients with type 2 diabetes mellitus may include hypertension, obesity, presence of dyslipidemia, effects of exogenous and endogenous insulin associated with the treatment of diabetes, degree of chronic hyperglycemia, duration of diabetes, and presence of hypoglycemia [7,8,11].

Sarcopenia is a risk factor for dementia in the elderly population without [12,13] or with [14] diabetes mellitus. Sarcopenia is a syndrome characterized by progressive and generalized loss of skeletal muscle mass and strength that is associated with the risk of physical dysfunction, poor quality of life, and death [15,16]. Elderly patients with diabetes exhibit a combination of impaired insulin secretion and increased insulin resistance, which might be due to a combination of adiposity and sarcopenia [14]. There are three criteria for sarcopenia diagnosis: low muscle mass, low muscle strength, and low physical performance [15,17]. However, it remains unclear how the diagnosis of sarcopenia and/or its three criteria are associated with the presence of MCI in patients with type 2 diabetes mellitus.

The main objectives of this study included assessing the clinical characteristics of type 2 diabetes patients with MCI and elucidating whether the diagnosis of sarcopenia and/or its criteria could act as its explanatory factors.

## 2. Methods

### 2.1. Study Design and Subjects

This is an observational retrospective cohort study. The study protocol was approved by the Fukushima Medical University Ethics Committee (Number 29118). Written informed consent was obtained from the patients recruited between January 2018 and December 2019 in the Department of Diabetes, Endocrinology and Metabolism, School of Medicine, Fukushima Medical University Hospital, Fukushima, Japan. This study was conducted according to the Ethical Guidelines for Medical and Health Research Involving Human Subjects enacted by Ministry of Health, Labour and Welfare (MHLW) of Japan [18,19]. Among 701 patients who gave written informed consent, 207 patients who did not have diabetes mellitus nor had type 1 diabetes mellitus were excluded from the study (Appendix A). The remaining 494 patients with type 2 diabetes mellitus were enrolled in the study and their paper and/or electrical medical records were obtained. After excluding for missing data, 438 patients were finally included for data analysis.

### 2.2. Measurements

#### 2.2.1. Patients’ Medical Records

Patient parameters, such as age, gender, history of diabetes mellitus, family and social history, medical checkup history, complications, medications, laboratory data, and all dates, were obtained from their paper and/or electrical medical records. Questionnaires were provided to record the data on smoking status (current smoker or not), drinking habits (everyday, sometimes, rarely, or never), regular exercise (exercise to sweat lightly for over 30 min on each occasion, two times weekly, walking >1 h/day, and fast walking), anti-hypertensive drug use, anti-hyperglycemic drug use, and lipid-lowering drug use. A participant was diagnosed with diabetes mellitus when fasting plasma glucose level was ≥126 mg/dL or the HbA1c level was ≥6.5% (48 mmol/mol), or if the participant regularly used anti-hyperglycemic drugs. A participant was diagnosed with hypertension if the systolic blood pressure was ≥140 mmHg or if the diastolic blood pressure was ≥90 mmHg, or if she/he regularly used antihypertensive drugs. A participant was diagnosed with dyslipidemia if high-density lipoprotein (HDL) cholesterol levels were <40 mg/dL (1.0 mmol/L), low-density lipoprotein (LDL) cholesterol levels were ≥140 mg/dL (3.6 mmol/L), or triglyceride levels were ≥150 mg/dL (1.7 mmol/L), or if they regularly used lipid-lowering drugs.

#### 2.2.2. Measurements of Parameters

Patients visited the clinic at 1–3 months intervals. Blood samples were collected at every visit in the morning after breakfast or after an overnight fasting for ≥10 h and were assayed within 1 h using automatic clinical chemical analyzers. The body weight, blood pressure, and pulse were measured at evety visit and the height, waist circumference, hand grip strength, body composition (muscle mass and fat mass), walking speed and MoCA-J (the Japanese version of the Montreal cognitive assessment) were measured annually by trained staff. Waist circumference was measured at the level of the umbilicus (cm) in the standing position. Hand grip strength (kg) was measured using an isokinetic dynamometer (Smedley hand dynamometer) on both hands, and values of the non-dominant arm were used. The compositions of fat and muscle in whole body, trunk, arms, and legs were assessed using a body composition analyzer (InBody 770, InBody Japan Inc., Tokyo, Japan) based on the segmental multi-frequency bioelectrical impedance analysis (SMF-BIA) [20,21]. The time required for walking 10 m was measured as described previously, with slight modifications [22,23]. In brief, we recorded the time required for walking 10 m by walking a straight path marked with a 10-m interval marked by start and end line tapes, by self-selected speeds without acceleration phase nor maximal phase, and measured by a hand-held stopwatch.

#### 2.2.3. Assessment of Mild Cognitive Impairment (MCI)

The Montreal Cognitive Assessment (MoCA), created in 1996 by Nasreddine, is a widely used screening assessment for detecting MCI [24]. The reliability and validity of the Japanese version of the MoCA (MoCA-J) had previously been tested in the Japanese population, and a cut-off point of 25/26 out of full 30 scores demonstrated a sensitivity of 93.0% and a specificity of 87.0% while screening for MCI [25]. Therefore, participants with MoCA-J score <26 were categorized into the MCI group and the participants with MoCA-J score ≥26 into the non-MCI group.

#### 2.2.4. Assessment of Sarcopenia

The definition and diagnosis of sarcopenia were based on Asian Working Group for Sarcopenia (AWGS): 2019 Consensus Update on Sarcopenia Diagnosis and Treatment [17]. In brief, “low muscle strength” was defined as handgrip strength <28 kg for men and <18 kg for women; the criterion for “low physical performance” was walking speed <1.0 m/s as evaluated by the time required for walking 10 m; “low appendicular skeletal muscle mass (ASM)” was defined as bioimpedance <7.0 kg/m² in men and <5.7 kg/m² in women. Sarcopenia was defined by low ASM and low muscle strength or low physical performance.

### 2.3. Statistical Analyses

A Kolmogorov Smirnov test was done to evaluated the normality of distribution. Parametric values were expressed as mean ± standard deviation (SD) and comparisons between two groups were made by two-tailed unpaired Student’s *t*-test. Non-parametric values were expressed by median (interquartile range 25%, 75%) and comparisons were made by a Mann–Whitney U-test. Categorical variables were shown as number (%) and were analyzed using the chi-square test. Univariate (Model 1) and multivariate-adjusted models were used to determine the independent contributors to MoCA-J <26 in Model 2 (sex, age, and body mass index (BMI)), Model 3 (sex, age, BMI, duration of diabetes mellitus, hypertension, dyslipidemia, smoking, drinking, estimated glomerular filtration rate (eGFR), and HbA1c), and Model 4 (sex, age, BMI, duration of diabetes mellitus, hypertension, dyslipidemia, coronary heart diseases, stroke, smoking, drinking, eGFR, and HbA1c). The odds ratios (ORs) between participant characteristics and MoCA-J <26 were calculated using logistic regression analysis. Values of *p* < 0.05 were considered as statistically significant. Statistical analyses were conducted using SPSS version 25 (SPSS, Inc., 233 South Wacker Drive, 11th Floor, Chicago, IL 60606-6307, USA).

## 3. Results

### 3.1. General Characteristics

Baseline characteristics of the patients are shown in Table 1. Among 438 participants with type 2 diabetes mellitus, 217 (49.5%) participants exhibited MoCA-J <26 (MCI group) and 221 (50.5%) participants exhibited MoCA-J ≥26 (non-MCI group). Patients of the MCI group were older, but the proportion of the male participants in this group was comparable to that in the non-MCI group. Systolic blood pressure between patients of both groups was comparable, but the diastolic blood pressure was lower in the MCI group. Body weight, BMI, waist circumference, as well as total fat mass were lower in the MCI group. Hand grip strength, walking speed, and skeletal mass index were all lower and the prevalence of sarcopenia was higher in the MCI group compared to the non-MCI group (13% vs. 4%). There were no significant differences in the levels of plasma glucose and HbA1c between the two groups; however, the levels of albumin, LDL-cholesterol, and eGFR were lower in the MCI group.

### 3.2. Unadjusted Odds Ratio

The unadjusted odds ratio (OR) for MCI (MoCA-J <26) is shown in Table 2. Diastolic blood pressure, body weight, BMI, fat mass, hand grip strength, walking speed, skeletal mass index, and levels of LDL-cholesterol and eGFR were inversely correlated and the risk of sarcopenia was directly correlated with MoCA-J <26. Age and the duration of diabetes mellitus were positively associated with MoCA-J <26. There were no associations of the levels of plasma glucose and HbA1c, but the lower levels of albumin, LDL-cholesterol, and eGFR were associated with MoCA-J <26.

### 3.3. Multivariate-Adjusted Odds Ratio

Multivariate-adjusted OR for MCI (MoCA-J <26) is shown in Table 3 and Figure 1. Overall, the non-adjusted OR of hand grip strength, walking speed, and skeletal mass index were negatively associated and that of sarcopenia was positively associated with MoCA-J <26 (Model 1). However, after adjusting for confounding factors (Models 2–4, see Table 3 footnote), only walking speed was significantly associated with MoCA-J <26.

## 4. Discussion

This study evaluated the clinical characteristics of the MCI group and the relationship between sarcopenia and/or its diagnostic criteria and MCI in Japanese patients with type 2 diabetes mellitus. We reported two major findings. First, we assessed the clinical characteristics of the MCI group in patients with type 2 diabetes mellitus. In patients of the MCI group, weight, BMI, waist circumference, and fat and muscle mass were lower compared to those of patients in the non-MCI group. Hand grip strength, walking speed, and skeletal mass index, which are associated with sarcopenia, were lower and the prevalence of sarcopenia was higher in the MCI group. Duration of diabetes mellitus was longer in the MCI group, but there was no significant difference between the two groups with respect to hypertension, dyslipidemia, and a history of coronary artery disease and stroke. There were no significant differences between the groups with respect to regular exercise habits, smoking history, and drinking history. Second, after correction for multivariate analysis, the OR for only the walking speed was found to be significant and the OR for other sarcopenia criteria and diagnosis of sarcopenia was not significant. The present study clarified that slow walking speed is the sole determinant criteria of sarcopenia of MCI in patients with type 2 diabetes mellitus. In patients with type 2 diabetes mellitus, it is necessary to verify whether the assessment of walking speed is effective for screening, early detection [26], and therapeutic intervention of MCI.

### 4.1. Characteristics of Patients in the MCI Group

Among the 438 patients with type 2 diabetes mellitus, 49.5% patients belonged to the MCI group and 50.5% to the non-MCI group. The age of patients in the MCI group was higher; however, there was no significant difference in the gender ratios of the two groups. Systolic blood pressure was comparable between both groups, but the diastolic blood pressure was lower in the MCI group. Body weight, BMI, abdominal circumference, as well as fat and muscle mass were lower in the MCI group. Hand grip strength, walking speed, and skeletal muscle index, which are the criteria for sarcopenia, were all lower, and the prevalence of sarcopenia was higher in the MCI group. Duration of diabetes mellitus was longer for patients in the MCI group, but there was no significant difference between the two groups with respect to hypertension, dyslipidemia, and a history of coronary artery disease and stroke. There was no significant difference in the habits of regular exercise, smoking, and drinking of the two groups.

The patients in the MCI group exhibited lower BMI, grip strength, muscle mass, fat mass, and walking speed, which were all consistent with the characteristics of sarcopenia [16,17]. Advanced age is a risk factor of MCI in subjects with or without diabetes mellitus [6,27]. Elderly patients with type 2 diabetes mellitus are likely to reflect a longer average duration of diabetes mellitus and a longer exposure to chronic hyperglycemia and glycemic excursions [7,8,11]. We observed longer durations of diabetes mellitus in patients of the MCI group, but no significant difference between the average A1c levels of both groups. Previous reports have shown a weak association between average A1c levels and MCI risk [28,29]. Another report showed that blood glucose fluctuations were more strongly associated with MCI risk than A1c level [30]. It has also been reported that risk of dementia is increased in patients with newly diagnosed diabetes mellitus [12], but not in patients with diabetes mellitus who are being treated [11]. In addition, findings from cross-sectional analyses on the association of HbA1c with cognitive function and cognitive decline in people with type 2 diabetes mellitus have been inconsistent [11]. Our results regarding HbA1C level agreed with those of previous studies [7,8,11].

### 4.2. Walking Speed and MCI

We examined the association between the diagnosis of sarcopenia and/or its criteria and MCI in Japanese patients with type 2 diabetes mellitus. On the criteria of AWGS [17], we defined sarcopenia by low ASM (skeletal mass index <7.0 kg/m² in men and <5.7 kg/m² in women) along with a low muscle strength (handgrip strength <28 kg for men and <18 kg for women) or low physical performance (walking speed <1.0 m/s). Interestingly, only walking speed (not muscle loss or weakness) is an independent determinant of MCI after adjusting for multiple factors, such as age, gender, BMI, duration of diabetes mellitus, hypertension, dyslipidemia, smoking, drinking, eGFR, HbA1c, and history of coronary heart diseases and stroke. Limited reports have evaluated the association between cognitive dysfunction and the criteria for sarcopenia in patients with type 2 diabetes mellitus. In Japanese patients with type 2 diabetes mellitus and cognitive dysfunction, the skeletal muscle strength and walking speed were lower [31,32]. These results were consistent with our results, and we also found that walking speed is the sole determinant for MCI among the three criteria for sarcopenia.

### 4.3. Why is Reduced Walking Speed Associated with MCI?

The mechanism of cognitive dysfunction in patients with diabetes mellitus has been suggested to be related to: (1) insulin resistance in the local brain [7,33]; (2) chronic inflammation, deposition of advanced glycation end-products (AGEs), and mitochondrial dysfunction; (3) τ (tau) deposition; (4) low adiponectin level in blood [34]; and (5) hypoglycemia due to exogenous and endogenous insulin [35,36]. Brain atrophy due to impaired glucose metabolism, including hypoglycemia, is suggested to be linked to the onset of MCI [37].

There are four possible mechanisms due to which walking speed could be linked to the onset of MCI. First, a slow walking speed might reflect an average decrease in physical activity. Patients who exercise regularly have a lower risk of Alzheimer’s disease and dementia [38,39]. However, there was no difference in the habits of regular exercise between our MCI and non-MCI groups. Second, a slow walking speed might indicate sarcopenia (muscle loss), decreased muscle quality, or reduced overall physical function. Patients with type 2 diabetes mellitus and visceral fat accumulation have been reported to have poor muscle quality [40]. Decreased skeletal muscle mass is likely to occur in patients with decreased insulin secretion. In such cases, muscle mass and quality might be reduced. Third, the walking function might reflect metabolism in the brain. Holtzer and colleagues suggested that elderly patients with diabetes mellitus exhibit altered frontal lobe function during walking and are at a risk of falling [41]. Fourth, it is possible that hypoglycemia caused by sulphonylurea and insulin is linked to abnormalities in brain metabolism and a reduced walking speed. However, in this study, walking speed was still a determinant of MCI, and was not related to hypoglycemia, even after adjusting the use of SU or insulin, which had a risk of hypoglycemia.

### 4.4. Walking Speed and MCI in the Subgroups

In subgroup analysis, we found that a group with age ≥60 years, male patients, and BMI <25 showed a significant OR of walking speed for MoCA-J <26 (Table 3 and Figure 1).

#### 4.4.1. Elderly vs. Young

Since the frequency of MCI was higher in the group ≥60 years than in the group <60 years (58% vs. 22%, χ² test *p* < 0.001), the sensitivity for detection of MCI may be higher for walking speed in the elderly than in the young.

#### 4.4.2. Non-Overweight Versus Overweight

Walking speed was a predictor for MCI only in the group with BMI <25, but not in the group with BMI ≥25. The non-overweight group had more sarcopenia (muscle loss), reduced muscle quality, or overall physical decline compared to the overweight group (Appendix A). It might be suggested that slower walking speed may be linked to the onset of MCI more closely in the non-overweight subgroup.

#### 4.4.3. Men vs. Women

Interestingly, the relationship between walking speed and MCI were different between men and women. There was no difference in median MoCA-J points between men and women. Although BMI was lower in men, values of sarcopenia components were higher in men: hand grip strength, muscle mass, skeletal muscle index and walking speed were all higher, and frequency of sarcopenia was lower. Although the frequency of high blood pressure and stroke history were comparable, median age and frequency of smoking and drinking history were higher in men. The ratio of prevalence of men to women differs on the subtype of dementia: females are at greater risk of developing Alzheimer’s disease dementia, elevated males are at greater risk of developing vascular dementia [42]. The impact of diabetes on dementia has been reported to be greater in women than men [43]. The current study cannot determine the reasons for gender difference in the relationship between walking speed and MCI. Sex differences in MCI subtypes in patients with diabetes mellitus may be considered to be one explanation in further studies.

### 4.5. Clinical Usefulness

The current study confirms that slow walking speed is a sole determinant criteria of sarcopenia of MCI in patients with type 2 diabetes mellitus. Moreover, we found that the relationship between walking speed and MCI were operative in the subgroups of age ≥60 years, male patients, and BMI <25. The results warrant us to verify whether the assessment of walking speed is effective for screening, early detection [24], and therapeutic intervention of MCI, especially for elderly, men and non-overweight subgroups.

### 4.6. Study Limitations

This study has several limitations. First, this study was done using a relatively small number of cases at a single hospital. However, our data can be considered as relatively valid because these were collected using standardized and uniform methods. Second, the data were limited to only the Japanese population and our results cannot be extended to other races. Third, this was a retrospective observational study and the cause of and resulting relationship between MCI and its risk factors could not be determined in this study.

## 5. Conclusions

This study showed that slow walking speed is a sole determinant criterion of sarcopenia of MCI in patients with type 2 diabetes mellitus. It was suggested that walking speed is an important factor, when considering the prediction and prevention of MCI development in patients with diabetes mellitus.

## Figures and Tables

**Figure 1 jcm-09-02133-f001:**
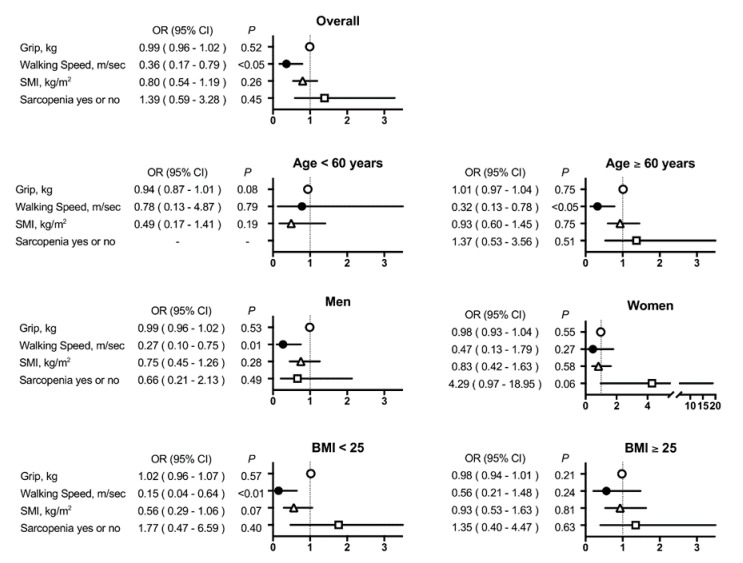
**Adjusted odds ratio (OR) for mild cognitive impairment (MCI) (Model 4).** ORs (5% confidential intervals) for MCI were calculated for hand grip strength (kg), walking speed (m/s), skeletal muscle index (kg/m²), and sarcopenia (yes or no) in overall patients with type 2 diabetes mellitus (*n* = 438), with respect to age <60 years or age ≥60 years, men or women, and BMI <25 or BMI ≥25. MCI was defined if MoCA-J (the Japanese version of Montreal Cognitive Assessment) score was <26. Model 4: adjusted for age, gender, BMI, duration of diabetes mellitus, hypertension, dyslipidemia, smoking, drinking, eGFR, HbA1c, and history of coronary heart diseases and stroke. Subgroup analysis (Table 3, lower panel) revealed that a group with age ≥60 years, male patients, and BMI <25 showed a significant OR of walking speed for MoCA-J <26. As shown in Appendix A, there was a difference in the MoCA-J values between patients aged <60 years and those aged ≥60 years (mean 27, interquartile range (26,29) vs. 25 (22,27) points, *p* < 0.01); however, no differences were observed between men and women and between patients with BMI <25 and those with BMI ≥25.

**Table 1 jcm-09-02133-t001:** Baseline characteristics of studied patients with type 2 diabetes mellitus.

	All Patients	MOCA-J Score ≥ 26	MOCA-J Score < 26	*p*-Value
Variables	*n* = 438	*n* = 221	*n* = 217	
Age, years	69 (60, 74)	63 (53, 70)	71 (66, 78)	<0.01
Male, *n* (%)	244 (56)	122 (55)	122 (56)	0.83
MOCA-J, points	26 (23, 27)	27 (27, 29)	23 (21, 25)	<0.01
**Anthropometry**				
Systolic blood pressure, mmHg	132 (120, 143)	130 (120, 142)	134 (121, 145)	0.20
Diastolic blood pressure, mmHg	74 ± 12	75 ± 12	72 ± 11	<0.05
Body weight, kg	65.4 (56.5, 78.0)	67.7 (59.3, 80.6)	61.8 (54.0, 75.0)	<0.01
Body mass index (BMI), kg/m²	25.1 (22.1, 29.3)	25.8 (22.4, 30.5)	24.7 (22.0, 28.2)	<0.05
Waist circumference, cm	92.3 (82.0, 100.7)	92.0 (82.3, 102.3)	90.0 (81.3, 98.8)	0.11
Fat mass, kg	20.9 (14.3, 29.0)	21.6 (14.5, 31.6)	19.8 (14.2, 26.7)	<0.05
Fat ratio, %	32.5 (25.1, 39.1)	33.1 (24.2, 40.6)	31.6 (25.4, 38.5)	0.50
Hand grip strength, kg	32.0 (25.0, 42.0)	34.0 (25.0, 43.5)	30.0 (23.0, 40.0)	<0.01
Walking speed, m/sec	1.54 (1.43, 1.82)	1.67 (1.43, 1.82)	1.54 (1.25, 1.67)	<0.01
Muscle mass, kg	42.7 (36.1, 48.9)	42.2 (37.8, 49.7)	41.6 (34.4, 47.0)	<0.01
Skeletal muscle index, kg/m²	7.1 (6.3, 7.9)	7.3 (6.6, 8.1)	6.9 (6.1, 7.8)	<0.01
Sarcopenia, *n* (%)	38 (9)	9 (4)	29 (13)	<0.01
**Comorbidities**				
Duration of diabetes mellitus, years	15 (9, 21)	12 (7, 18)	16 (11, 23)	<0.01
Hypertension, *n* (%)	304 (69)	150 (68)	154 (71)	0.48
Dyslipidemia, *n* (%)	302 (69)	156 (71)	146 (67)	0.41
Coronary heart diseases, *n* (%)	64 (15)	30 (14)	34 (10)	0.54
Stroke, *n* (%)	36 (9)	13 (6)	23 (11)	0.07
**Life habits**				
Regular walking, *n* (%)	173 (40)	87 (40)	86 (40)	0.99
Current or ex-smoker, *n* (%)	231 (53)	128 (58)	103 (48)	0.11
Current or ex-drinker, *n* (%)	195 (45)	105 (48)	90 (42)	0.52
**Blood measurements**				
Albumin, g/dL	4.3 (4.0, 4.4)	4.3 (4.1, 4.5)	4.2 (4.0, 4.4)	<0.01
AST, U/L	21 (17, 28)	21 (18, 28)	21 (17, 28)	0.41
ALT, U/L	19 (14, 29)	20 (14, 32)	18 (13, 27)	<0.01
γGT, U/L	24 (17, 40)	25 (18, 44)	23 (16, 38)	0.11
Fasting plasma Glucose, mg/dL	132 (118, 153)	131 (116, 150)	134 (121, 157)	<0.05
HbA1c, %	6.9 (6.4, 7.5)	6.8 (6.4, 7.4)	7.0 (6.5, 7.5)	0.17
LDL cholesterol, mg/dL	101 (82, 118)	104 (84, 125)	99 (81, 116)	<0.05
HDL cholesterol, mg/dL	54 (46, 64)	54 (47, 64)	53 (44, 63)	0.09
Triglycerides, mg/dL	103 (73, 153)	107 (70, 162)	101 (74, 143)	0.34
Creatinine, mg/dl	0.82 (0.69, 0.99)	0.79 (0.68, 0.97)	0.88 (0.73, 1.02)	<0.01
eGFR, ml/min/1.73 m²	64.3 ± 18.6	67.9 ± 18.7	60.6 ± 17.7	<0.01
**Medications**				
Sulfonylurea, *n* (%)	43 (10)	19 (9)	24 (11)	0.39
Insulin, *n* (%)	125 (29)	60 (27)	65 (30)	0.52
Biguanide, *n* (%)	224 (51)	103 (48)	121 (55)	0.13
Glinide, *n* (%)	113 (26)	58 (27)	55 (25)	0.66
Pioglitazone, *n* (%)	138 (32)	66 (30)	72 (33)	0.63
α-Glucosidase Inhibitor, *n* (%)	87 (20)	47 (22)	40 (18)	0.35
Dipeptidyl peptidase-4 inhibitors, *n* (%)	271 (62)	141 (65)	130 (59)	0.19
Glucagon-like peptide 1 receptor agonist, *n* (%)	33 (8)	14 (7)	19 (9)	0.40
Sodium–glucose cotransporter (SGLT) inhibitors, *n* (%)	95 (22)	44 (20)	51 (23)	0.48

Median (25%, 75%), Mean ± SD or number (%). MoCA-J: the Japanese version of the Montreal cognitive assessment; Aspartate aminotransferase AST: alanine aminotransferase; γGT: γ-glutamyl transpeptidase; LDL: low density lipo protein; HDL: high density lipo protein; eGFR: estimated glomerular filtration rate.

**Table 2 jcm-09-02133-t002:** Unadjusted odds ration for mild cognitive impairment (MCI) in patients with type 2 diabetes mellitus.

Variables	Unadjusted Odds Ratio (95% CI)	*p*-Value
Age, years	1.08 (1.06–1.11)	<0.01
Male, yes or no	1.04 (0.72–1.52)	0.83
**Anthropometry**		
Systolic blood pressure, mmHg	1.01 (0.99–1.02)	0.38
Diastolic blood pressure, mmHg	0.98 (0.96–1.00)	<0.05
Body weight, kg	0.98 (0.97–0.99)	<0.01
BMI, kg/m^2^	0.96 (0.93–0.99)	<0.05
Waist, cm	0.99 (0.98–1.00)	0.09
Fat mass, kg	0.98 (0.97–1.00)	<0.05
Fat ratio, %	0.99 (0.98–1.01)	0.50
Hand grip strength, kg	0.97 (0.95–0.99)	<0.01
Walking speed, m/sec	0.16 (0.08–0.31)	<0.01
Muscle mass, kg	0.95 (0.93–0.98)	<0.01
Skeletal muscle index, kg/m^2^	0.73 (0.61–0.86)	<0.01
Sarcopenia, yes or no	3.63 (1.68–7.87)	<0.01
**Comorbidities**		
Duration of diabetes mellitus, years	1.05 (1.03–1.07)	<0.01
Hypertension, yes or no	1.16 (0.77–1.74)	0.48
Dyslipidemia, yes or no	0.84 (0.56–1.27)	0.41
Coronary heart diseases, n (%)	1.18 (0.70–2.01)	0.54
Stroke, n (%)	1.90 (0.94–3.85)	0.08
**Life habits**		
Regular walking, yes or no	1.00 (0.68–1.46)	0.99
Current or ex-smoker, yes or no	0.85 (0.69–1.04)	0.11
Current or ex-drinker, yes or no	0.92 (0.71–1.19)	0.52
**Blood measurements**		
Albumin, g/dL	0.37 (0.21–0.66)	<0.01
AST, U/L	0.99 (0.98–1.01)	0.32
ALT, U/L	0.98 (0.97–1.00)	<0.01
γGT, U/L	1.00 (1.00 - 1.00)	0.58
Fasting plasma Glucose, mg/dL	1.00 (1.00–1.01)	0.15
HbA1c, %	1.09 (0.90–1.21)	0.39
LDLcholesterol, mg/dL	0.99 (0.99–1.00)	<0.05
HDL cholesterol, mg/dL	0.99 (0.97–1.00)	0.09
Triglycerides, mg/dL	1.00 (1.00–1.00)	0.74
Creatinine, mg/dl	1.21 (0.82–1.79)	0.34
eGFR, ml/min/1.73 m^2^	0.98 (0.97–0.99)	<0.01
**Medications**		
Sulfonylurea, yes or no	1.32 (0.70–2.49)	0.39
Insulin, yes or no	1.15 (0.76–1.74)	0.52
Biguanide, yes or no	0.75 (0.51–1.09)	0.13
Glinide, yes or no	1.10 (0.72–1.69)	0.66
Pioglitazone, yes or no	0.91 (0.60–1.35)	0.63
α-Glucosidase Inhibitor, yes or no	1.25 (0.78–2.00)	0.35
Dipeptidyl peptidase-4 inhibitors, yes or no	1.30 (0.88–1.91)	0.19
Glucagon-like peptide 1 receptor agonist, yes or no	0.73 (0.36–1.50)	0.40
Sodium–glucose cotransporter (SGLT) inhibitors, yes or no	0.85 (0.54–1.34)	0.48

**Table 3 jcm-09-02133-t003:** Multivariate-adjusted odds ratio for mild cognitive impairment (MCI) in patients with type 2 diabetes mellitus odds ratio for MCI.

	Model 1 (Unadjusted)	*p*	Model 2	*p*		Model 3		*p*		Model 4		*p*
	OR (95% CI)		OR (95% CI)			OR (95% CI)				OR (95% CI)		
**Overall**												
Hand grip strength, kg	0.97 (0.95–0.99)	<0.01	0.99 (0.96–1.02)	0.49		0.99 (0.96–1.02)		0.46		0.99 (0.96–1.02)		0.52
Walking Speed, m/sec	0.16 (0.08–0.31)	<0.01	0.34 (0.16–0.72)	<0.01		0.35 (0.16–0.75)		<0.01		0.36 (0.17–0.79)		<0.05
Skeletal muscle index, kg/m²	0.73 (0.61–0.86)	<0.01	0.76 (0.52–1.11)	0.15		0.78 (0.52–1.15)		0.20		0.80 (0.54–1.19)		0.26
Sarcopenia yes or no	3.63 (1.68–7.87)	<0.01	1.51 (0.65–3.47)	0.34		1.46 (0.62–3.43)		0.39		1.39 (0.59–3.28)		0.45
**Age < 60 yearls old**												
Hand grip strength, kg	0.99 (0.95–1.03)	0.54	0.94 (0.88–1.00)	0.07	0.93 (0.87–1.00)		0.06		0.94 (0.87–1.01)			0.08
Walking Speed, m/sec	0.63 (0.13–3.02)	0.56	0.67 (0.13–3.56)	0.64	0.69 (0.12–4.12)		0.68		0.78 (0.13–4.87)			0.79
Skeletal muscle index, kg/m²	1.00 (0.68–1.49)	0.98	0.45 (0.18–1.15)	0.1	0.40 (0.14–1.10)		0.08		0.49 (0.17–1.41)			0.19
Sarcopenia yes or no	-	-	-	-	-		-		-			-
**Age ≥ 60 yearls old**												
Hand grip strength, kg	0.98 (0.96–1.00)	<0.05	1.01 (0.97–1.04)	0.76	1.01 (0.97–1.04)		0.79		1.01 (0.97–1.04)		0.75	
Walking Speed, m/sec	0.16 (0.08–0.35)	<0.01	0.31 (0.13–0.74)	<0.01	0.31 (0.13–0.76)		<0.05		0.32 (0.13–0.78)		<0.05	
Skeletal muscle index, kg/m²	0.83 (0.67–1.02)	0.08	0.89 (0.58–1.34)	0.57	0.91 (0.59–1.42)		0.68		0.93 (0.60–1.45)		0.75	
Sarcopenia yes or no	3.26 (1.38–7.68)	<0.01	1.49 (0.59–3.77)	0.4	1.41 (0.55–3.65)		0.47		1.37 (0.53–3.56)		0.51	
**Men**												
Hand grip strength, kg	0.95 (0.92–0.98)	<0.01	0.99 (0.96–1.02)	0.44		0.99 (0.95–1.02)		0.44		0.99 (0.96–1.02)	0.53	
Walking Speed, m/sec	0.13 (0.05–0.32)	<0.01	0.26 (0.10–0.69)	<0.01		0.25 (0.09–0.69)		<0.01		0.27 (0.10–0.75)	0.01	
Skeletal muscle index, kg/m²	0.68 (0.51–0.89)	<0.01	0.73 (0.45–1.18)	0.2		0.71 (0.43–1.18)		0.19		0.75 (0.45–1.26)	0.28	
Sarcopenia yes or no	1.73 (0.61–4.91)	0.31	0.72 (0.24–2.19)	0.56		0.73 (0.23–2.29)		0.59		0.66 (0.21–2.13)	0.49	
**Women**												
Hand grip strength, kg	0.92 (0.88–0.97)	<0.01	0.99 (0.94–1.05)	0.75	0.98 (0.93–1.04)		0.52		0.98 (0.93–1.04)		0.55	
Walking Speed, m/sec	0.18 (0.07–0.50)	<0.01	0.49 (0.15–1.63)	0.25	0.47 (0.13–1.71)		0.25		0.47 (0.13–1.79)		0.27	
Skeletal muscle index, kg/m²	0.63 (0.47–0.84)	<0.01	0.82 (0.44–1.52)	0.52	0.81 (0.42–1.59)		0.55		0.83 (0.42–1.63)		0.58	
Sarcopenia yes or no	8.00 (2.28–28.0)	<0.01	3.51 (0.90–13.7)	0.07	4.33 (0.99–19.0)		0.05		4.29 (0.97–19.0)		0.06	
**BMI < 25**												
Hand grip strength, kg	0.97 (0.95–1.00)	0.07	1.01 (0.96–1.05)	0.83	1.01 (0.96–1.06)		0.84		1.02 (0.96–1.07)		0.57	
Walking Speed, m/sec	0.07 (0.02–0.24)	<0.01	0.14 (0.04–0.52)	<0.01	0.13 (0.03–0.52)		<0.01		0.15 (0.04–0.64)		<0.01	
Skeletal muscle index, kg/m²	0.67 (0.48–0.95)	<0.05	0.57 (0.31–1.04)	0.07	0.51 (0.27–0.95)		<0.05		0.56 (0.29–1.06)		0.07	
Sarcopenia yes or no	3.51 (1.12–11.1)	<0.05	1.64 (0.48–5.58)	0.43	1.91 (0.52–6.98)		0.33		1.77 (0.47–6.59)		0.4	
**BMI ≥ 25**												
Hand grip strength, kg	0.97 (0.95–0.99)	<0.01	0.98 (0.95–1.02)	0.28	0.98 (0.94–1.01)		0.22		0.98 (0.94–1.01)		0.21	
Walking Speed, m/sec	0.23 (0.10–0.52)	<0.01	0.55 (0.21–1.40)	0.21	0.56 (0.21–1.47)		0.24		0.56 (0.21–1.48)		0.24	
Skeletal muscle index, kg/m²	0.75 (0.60–0.95)	<0.05	0.96 (0.57–1.61)	0.88	0.94 (0.54–1.63)		0.81		0.93 (0.53–1.63)		0.81	
Sarcopenia yes or no	3.55 (1.24–10.2)	<0.05	1.36 (0.43–4.33)	0.61	1.35 (0.41–4.41)		0.62		1.35 (0.40–4.47)		0.63	

Model 1 (Unadjusted); Model 2 (sex, age, and BMI); Model 3 (sex, age, BMI, duration of diabetes mellitus, hypertention, dyslipidemia, smoking, drinking, eGFR, and HbA1c); Model 4 (sex, age, BMI, duration of diabetes mellitus, hypertention, dyslipidemia, coronary heart diseases, stroke, smoking, drinking, eGFR, HbA1c).

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
