# Peer review of "Walking Speed is the Sole Determinant Criterion of Sarcopenia of Mild Cognitive Impairment in Japanese Elderly Patients with Type 2 Diabetes Mellitus"

_jcm, 2020, doi:10.3390/jcm9072133_

Round 1
Reviewer 1 Report
Dear Authors,
It was my pleasure to read the manuscript written by Noritaka Machii, Akihiro Kudo, Haruka Saito, et all, titled Walking speed is the sole determinant of mild 2 cognitive impairment in Japanese patients with type 3 2 diabetes mellitus. I have read the manuscript with the great interest.
This is very interesting work, very carefully planned and conducted. The manuscript is presented in very careful way, all selected paragraphs are constructed properly. The experimental design is good and I think that the work suits the objectives of Journal of
Clinical Medicine. Nevertheless, I would like to address a few comments:
Introduction:
No remarks
Materials and Methods
Line 94-95: Please move unit (cm) just after the measured body dimension (waist circumference).
Results:
It the current version of the manuscript the normality of distribution was not checked and it was assumed that continuous variables will be described as mean +/- standard deviation, then results were compared by Student's t test.
The distribution of normality should be properly tested and presented as Me (Q1-Q3, as oblique distributions) and then nonparametric tests should be used for connected analyses.
For the readers convenience the results should be also presented in figures, since the manuscript contains only tables which, in the current form, are difficult to read.
Discussion
Lines 224-225: Please indicate to which previous studies you refer to, as this sentence is ambiguous.
Summary:
The manuscript is well planned and executed. The text is coherent and it reads well. The authors are aware of the limitation of the study.
My main concern is connected with statistical analysis of the results and presentation of the results.
My second request would be to improve the manuscript in the two above-mentioned places and to adjust citation style to journal requirements, as right now the authors use either brackets or parentheses.
Final recommendation:
Resubmit after major revision
Reviewer 2 Report
The manuscript addresses an interesting topic, but there are some minor issues which should be addressed in a revised version of the manuscript.
1.- The title should be adressed. I propose: Walking speed is the sole determinant criteria of sarcopenia of mild cognitive impairment in Japanese elderly patients with type 2 diabetes mellitus
2.- Please use people first language (persons with type 2 diabetes instead of diabetic persons).
3.- Please choose between type 2 diabetes or type 2 diabetes mellitus and write it throughout the text
4.- The conclusions should be modified: "This is the first study" should be deleted
5.- After line 49, new paragraph
6.- The measurements section should be improved. There is no order making reading difficult
7.- Line 100, which slight modifications?
8.- Line 120, There is a normal distribution? KOLMOGOROV SMIRNOV test shoud be used.
9.- Line 129, information in parentheses shoud be adressed
10.- The results part is rather long. Please do not repeat the whole figures and tables in the main text. The results part would benefit from shortening.
11.- Table 1, waist circumference. In addition, after medications n (%). The same for supplement 2.
12.- Table 3, lowercase p and italic
13.- Figure 1, why authors use 65 years? At the moment the UN agreed cutoff is 60+ years to refer to the older population.
14.- The Discussion section should have a paragraph on how the variables of elderly subjects are discussed. Highlight this information in the conclusion section. The differences among sexes and owerweight should be also explained.
15.- Line 242, AGEs
16.- Authors should comment on the clinical usefulness of their results. For example sceening elderly patients using walking speed.
17.- Supplement 1, there is missing (n=)
Round 2
Reviewer 1 Report
Dear Authors,
Thank you for your work and current presentation of the manuscript.
I would like to congratulate you very interesting study.
Author Response
Thank you for valuable advices. We are pleased the current manuscript are looking pretty good.